# A Potential Role of Plant/Macrofungi/Algae-Derived Non-Starch Polysaccharide in Colitis Curing: Review of Possible Mechanisms of Action

**DOI:** 10.3390/molecules27196467

**Published:** 2022-10-01

**Authors:** Jinxiu Feng, Jingzhang Geng, Jinhui Wu, Huiying Wang, Yanfei Liu, Bin Du, Yuedong Yang, Haitao Xiao

**Affiliations:** 1Hebei Key Laboratory of Natural Products Activity Components and Function, Hebei Normal University of Science and Technology, Qinhuangdao 066004, China; 2College of Biological Science and Engineering, Shaanxi University of Technology, 1 East 1st Ring Road, Hanzhong 723000, China; 3School of Pharmaceutical Sciences, Health Science Center, Shenzhen University, Shenzhen 518060, China

**Keywords:** plant/macrofungi-derived polysaccharide, inflammatory bowel disease, mechanisms of action

## Abstract

Multiple in vitro and in vivo model investigations have suggested a broad spectrum of potential mechanisms by which plant/macrofungi-derived non-starch polysaccharides may play a role in the treatment of inflammatory bowel disease (IBD). This article reviews the in vivo and in vitro evidence of different plant-derived polysaccharides for IBD therapy. Their underlying mechanisms, particularly the molecular mechanisms associated with protective effects in the treatment and prevention of IDB, have been well summarized, including anti-inflammatory, epithelial barrier repair, and the regulation of intestinal flora. Emerging studies have observed the potent role of probiotics in IBD, particularly its ability to modulate gut microbiota, a well-known key factor for IBD. In summary, plant/macrofungi-derived polysaccharides have the potential to be a promising agent for the adjuvant treatment and prevention of IBD and will contribute to the design of well-designed clinical intervention trials that will ultimately improve the therapy of IBD.

## 1. Introduction

Over the past few decades, the incidence of inflammatory bowel disease (IBD) has increased significantly. As a natural product, polysaccharides are an ideal target for selection. IBD is a chronic recurrent inflammatory disease of the gastrointestinal tract that affects the ileum, rectum, and colon and is caused by a variety of genetic and environmental factors. Its clinical manifestations are diarrhea, abdominal pain, and even bloody stools. The disease includes ulcerative colitis (UC) and Crohn’s disease (CD) [1]. Ulcerative colitis is a continuous inflammation of the colonic mucosa and submucosa. The disease usually first affects the rectum and gradually spreads to the entire colon. Crohn’s disease can affect the entire digestive tract. It is a discontinuous full-thickness inflammation. The most commonly affected sites are the terminal ileum, colon, and perianal [2]. Over the past decade, IBD has become a global public health challenge. Young people are the major group affected by IBD as a chronic disease [3]. Patients with IBD are usually treated with aminosalicylic acid [4,5], corticosteroids [6], immunosuppressants, antibiotics [7], and biologics [8]. However, these drugs are expensive and cannot completely cure colitis. In addition, most patients eventually develop immune tolerance to these drugs, and the side effects associated with the use of these drugs are quite extensive, and some are even life-threatening [8]. As a result, medical scientists are working to develop a new adjuvant treatment strategy, especially in the early stages of IBD, which may involve natural products of food origin, as dietary changes have shown the potential to help induce disease relief [9]. As a natural product, polysaccharides are an ideal target for selection. Many different studies have proved this in vivo and in vitro.

Plant/macrofungi/algae-derived non-starch polysaccharides from various sources, including grains, mushrooms, plants, etc., have many biological activities. Polysaccharides composed of different monosaccharides are called heteropolysaccharides, such as gum arabic, which is composed of pentose and galactose. The polysaccharide is not a pure chemical substance, but a mixture of substances with different degrees of polymerization. The structural unit of a polysaccharide is a monosaccharide, and the relative molecular mass of a polysaccharide is from tens to tens of millions Da. The structural units are connected by glycosidic bonds. Common glycosidic bonds include α-1,4-, β-1,4-, and α-1,6-glycoside bonds. Structural units can be linked into straight chains or branched chains. Linear chains are generally linked by α-1,4-glycosidic bonds (such as starch) and β-1,4-glycosidic bonds (such as cellulose). The point of attachment of the chain is often an alpha-1,6-glycosidic bond. Polysaccharides have excellent clinical effects in immune regulation, antiviral and anticancer, and pharmaceuticals. Since polysaccharides have significant pharmacological activities and few side effects, they have received extensive attention in recent years [10,11]. The purpose of this study is to review the literature on the effects of polysaccharides on IBD treatment and possible mechanisms of action.

## 2. Improvement of Inflammatory

Available evidence indicates that dysfunctions of innate and adaptive immune regulation lead to abnormal intestinal inflammatory responses in patients with IBD. The immunomodulatory effect of polysaccharides on IBD from pharmacological and clinical studies, referred to as biological response modifiers (BRMs), is one of the most active areas in polysaccharide research. The primary effect of polysaccharides is to enhance and/or activate immune cell responses, including anti-oxidative stress, enhancing the secretion of cytokines and chemokines, as well as the production of short-chain fatty acids (SCFAs) accompanied by inflammation.

Extensive studies demonstrate that astragalus polysaccharide (APS), a Chinese medicine widely used to enhance immunity, exhibits potential beneficial effects to alleviate the severity and colonic inflammation of colitis by up-regulating anti-inflammatory factors and down-regulating pro-inflammatory factors. In TNBS-induced experimental colitis in rats, the administration of APS ameliorated colitis by restoring the number of T regulatory cells (Treg) and inhibiting IL-17 levels in Peyer’s patches [12]. This balancing mechanism of APS on Treg cells and T helper cells 17 (Th-17) was found to adjust the GATA-3/T-bet ratio to drive T helper cells 1 (Th1) to T helper cells 2 (Th2) [13]. In the same model, the mRNA expression and protein production of NFATc4 increased, while TNF-α and IL-1β expressions (both mRNA and protein) were downregulated by this treatment, thereby preventing weight loss caused by TNBS-induced colitis and improving macroscopic and microscopic scores [14]. Additionally, a study carried out in DSS-induced colitis provided reliable evidence that APS attenuates murine colitis through the inhibition of the NOD-like receptor protein 3 (NLRP3) inflammasome, which acts to reduce the production of inflammatory cytokines such as interleukin-18 and interleukin-1β [15]. A recent study revealed that wild peony polysaccharides also improved DSS-induced colitis, involving the regulation of Th1/Th2 and Th17/Treg balance [16].

*Rheum tanguticum* polysaccharide (RTP) has been used as a remedy for gastrointestinal diseases for more than two thousand years in China. It was reported that mice that received RTP at doses of 200 mg/kg per day significantly ameliorated TNBS-induced colonic damage, which was associated with suppression of NF-κB [17,18]. Furthermore, Liu et al. investigated the protective effects of RTP on TNBS-induced colitis in rats and reported that the rectal administration of RTP effectively attenuated the severity of TNBS-induced colitis with the modulation of CD4+T cell dysfunction [19,20]. Moreover, Th1/Th2 cytokine production balance has been shown to be one of the mechanisms by which RTP effectively inhibits inflammation [18,19,20]. Additionally, targeting the mannose receptor in macrophages and down-regulation of Th1-polarized immune responses, decreasing cell survival and SOD activity, and increasing production of MDA, LDH leakage and cell apoptosis can also be possible mechanisms [20,21].

*Ganoderma lucidum* polysaccharides (GLP) were reported to prevent inflammation, maintain intestinal homeostasis, and regulate intestinal immunological barrier functions in mice with DSS-induced colitis. Through the suppression of the immune responses, including decreased secretion of proinflammatory cytokines, such as TNF-α, IL-6, IL-1β, and IL-17A, it increased the populations of B cells and decreased the populations of Th17 cells, NK cells, and NKT cells [22]. In a model of indomethacin-induced colitis, GLP also displayed anti-inflammatory effects through the granulocyte-macrophage colony-stimulating factor in a small intestinal injury [23].

Oxidative stress signaling is involved in and promotes the development of IBD through multiple levels of function. Accumulating data obtained from both experimental models and clinical studies has highlighted the beneficial roles of polysaccharides in treating IBD, such as antioxidant and anti-apoptotic activity [24]. A previous study showed that a pectic polysaccharide of a common cranberry protected a colitic rat from acetic acid (AA)-induced injury by exerting a reduction of neutrophil infiltration and antioxidants [25]. The oral intake of a polysaccharide extracted from *Hericium*
*erinaceus* (HE) can reduce intestinal inflammation in a DSS-stimulated colitis model by adjusting the production of NO, MDA, T-SOD, and MPO to repair the oxidative damage of the mucosal barrier, and down-regulating the expression of COX-2, iNOS, and cytokines via blocking NF-kB, MAPK, and PI3K/AKT signaling pathways [26]. Wang et al. investigated the protective effect of polysaccharides from mycelium of HE against acetic acid-induced colitis in rats and found that the polysaccharide exhibited an excellent antioxidant capacity both in vivo and in vitro. It is also involved in preventing ROS damage to mitochondria by increasing oxygen consumption and eliminating the ROS substrate [27]. Dietary supplementation of *Angelica sinensis* polysaccharides (ASP) (5 mg/mL and 10 mg/mL) in a TNBS-induced colitis model significantly reduced body weight and glutathione (GSH) content, increased malondialdehyde concentration and raised the amount of apoptotic cells [28]. Additionally, ASP has a protective effect on immunological colon injury induced by TNBS in rats, which manifested as significant increases in MPO activity, NO contents, as well as the levels of TNF-α and IL-2 in colonic tissues, accompanied by a reduction in colonic TGF-β protein expression, SOD activity, and IL-10 [29]. Antioxidant properties of polysaccharides derived from *Morinda citrifolia* Linn and konjac oligosaccharide act against intestinal damage by reducing MPO activity and levels of GSH, MDA, NO3/NO2, pro-inflammatory cytokines, and COX-2 expression [1,20]. Furthermore, *Angelica sinensis* Polysaccharide could protect against UC through the synergistic effect of inhibiting the expression of proinflammatory cytokines, decreasing the IEC apoptosis [30]. These results show that a better understanding of the role of oxidative stress in IBD will absolutely contribute to improved treatment of IBD, especially in the combined medication plan involving the use of natural and synthetic antioxidant compounds.

The balance between pro-inflammatory and anti-inflammatory factors is precisely regulated in the GI to maintain intestinal homeostasis [31]. Polysaccharides can regulate colitis by up-regulating inflammatory factors and down-regulating inflammatory factors. Liu et al. indicated that Oat Beta-Glucan ameliorates DSS-induced ulcerative colitis at doses of 500 and 1000 mg/kg per day through the inhibition of the expression of pro-inflammatory factors [32]. In another study, Oat Beta-Glucan had beneficial effects in TNBS-induced colitis due to an inhibition of mucosa and submucosa lymphocyte infiltration [33]. Moreover, *Arctium lappa* ameliorated the dysregulation of pro-inflammatory cytokines (IL-1β, IL-6, and TNF-α) and anti-inflammatory cytokine (IL-10) [34]. It has been reported that the treatment of saponins and polysaccharides from *Codonopsis pilosula* Nannf could significantly inhibit DSS-induced colitis, and the expression of anti-inflammatory cytokines was upregulated when the secretion of proinflammatory cytokines correlated with Th17/Treg was downregulated, as well as enhanced the production of short-chain fatty acids (SCFA) [35]. In addition, Li et al. found that an alkali-soluble polysaccharide from purple sweet potato could ameliorate the damage to the mucosal barrier via downregulating the expression of pro-inflammatory cytokines [36]. Similar protective effects were found in the application of carboxymethyl polysaccharide against IBD in mice by decreasing the levels of pro-inflammatory cytokines and increasing the levels of anti-inflammatory cytokines [37]. An in vivo study showed that the anti-colitic effect of polysaccharide, extracted from *Scutellaria baicalensis* Georgi, on AA-induced colitis was found through the suppressing NF-κB signaling and the NLRP3 inflammasome [38]. In parallel, *Cynanchum wilfordii* polysaccharides are efficient in DSS-induced colitis by inhibiting NF-κB activation [39]. Subsequently, the rats orally receiving Selenium nanoparticles decorated with *Ulva lactuca* polysaccharides could effectively attenuate colitis by inhibiting NF-κB-mediated hyper inflammation in DSS-induced colitis rats and human THP-1 cells [40]. Additionally, the in vivo and in vitro study of polysaccharides from *Lycium barbarum* and *Astragalus membranaceusat* (a ratio of 2:3) on ulcerative colitis elucidated the potential mechanism of action in epithelial cell proliferation through blocking the JAK2/STAT-3 signaling pathway [41]. Lin et al. investigated the anti-inflammatory effect of aloe polysaccharide in LPS-stimulated NCM460 cells and found that aloe polysaccharide could significantly reduce LPS-induced IL-6 expression and control the apoptosis of colonic tissues by inhibiting the NLRP3 inflammasome and β-arrestin1 signaling pathways [42]. A study from Jin-Hua Tao and his co-workers reported that *Chrysanthemum* polysaccharides exhibit excellent anti-inflammatory effects in the attenuation of overall clinical scores and various pathological markers of TNBS-induced colitis via the regulation of the metabolic profiling and NF-κB/TLR4 and IL-6/JAK2/STAT3 signaling pathways. Hu et al. investigated the anticolitic effects of polysaccharides isolated from *Phellinus linteus* mycelia (PLP) on DSS-induced colitis in rats. It was found that rats receiving PLP significantly improved the health status of mice and suppressed DSS-induced pathological alterations with the reduction of inflammatory cytokine expressions via MAPK and PPAR signaling pathways [43]. In another study, a polysaccharide from *Flammuliana velutipes* (FVP) improved DSS-induced colitis via the regulation of colonic microbial dysbiosis and inflammatory responses by blocking the TLR4-NF-κB signal pathway, indicating that FVP is a potent agent for treating colitis [44]. The possible mechanism of polysaccharides treat colitis in three aspects are shown in Figure 1 and Table 1.

## 3. Recovery of Epithelial Barrier Function

The intestinal epithelium forms a physical barrier with a selective passage function, that can control the absorption of nutrients, water, and electrolytes, prevent the passage of toxins, microorganisms, and foreign antigens, and play the basic function of maintaining intestinal balance [56,57]. Tight junctions (TJs), which are mediated by proteins such as claudins, occludin, and zonula occludens, are necessary for epithelial barrier maintenance [58]. Therefore, the destruction of the intestinal barrier’s function can increase the permeability of epithelial cells, thereby inducing inflammation and breaking the immune balance. The current medical treatment of IBD relies on anti-inflammatory drugs for a long time, which will cause serious side effects such as secondary infection and immunosuppression; therefore, new therapeutic targets need to be explored. Promoting and protecting the integrity of the intestinal barrier may be a way to treat IBD [59].

It has been reported that rhamnogalacturonan can ameliorate intestinal barrier function in DSS-induced colitis, which was associated with the protection of the colon epithelium, the promotion of the maintenance of mucosal enterocytes and mucus-secreting goblet cells, the conservation of collagen homeostasis, and an increase in cell proliferation. In addition, it reduced the cellular permeability after exposure to IL-1β, while decreasing IL-8 secretion and claudin-1 expression and preserving the distribution of occludin. Furthermore, RGal accelerated wound healing in the Caco-2 epithelial cell line [60]. In the same model, lachnum polysaccharide was shown to be a potential natural agent for protecting mice from IBD and it restored intestinal barrier integrity by regulating the expression of tight junction proteins and mucus layer protecting proteins [61]. Additionally, in DSS-induced colitis, Angelica sinensis polysaccharide effectively alleviated the symptoms of ulcerative colitis (UC) in rats. This is because Angelica sinensis polysaccharide can suppress the expression of the proinflammatory cytokines (interleukin [IL]-6, IL-1b, and tumor necrosis factor-alpha), improve the expressions of tight junction proteins, such as zona occludens 1, occludin, and claudin-1, and reduce apoptosis [30].

Wild jujube polysaccharides protect against IBD by enabling enhanced intestinal barrier function involving the activation of AMPK. It suppressed the inflammatory response via the attenuation of TNF-α, IL-1β, IL-6, and MPO activity in colitis rats in TNBS-induced colitis. Furthermore, in the Caco-2 cell, the author demonstrated that the alleviation of colon injury by wild jujube polysaccharides was associated with a barrier function by the assembly of tight junction proteins [52].

In one study, Scutellaria baicalensis Georgi polysaccharide ameliorated colonic pathological damage and decreased MPO activity of ulcerative colitis mice induced by DSS. Moreover, the intestinal barrier was repaired due to the up-regulated expressions of ZO-1, Occludin, and Claudin-5. SP2-1 remarkably enhanced the levels of acetic acid, propionic acid, and butyric acid in DSS-treated mice [62].

## 4. Regulation of Gut Microbiota

The human gastrointestinal (GI) tract is populated by a diverse, highly mutualistic microbial flora, which is known as the microbiome. Different microorganisms colonizing the digestive tract constitute the human intestinal microbiome, which is involved in host digestion, energy metabolism, immune response regulation, and protection of the digestive tract from harmful pathogens. Disruptions to the microbiome in the GI tract, often referred to as dysbiosis, are associated with IBD [63]. A healthy gut environment is regulated by the exquisite balance of intestinal microbiota, metabolites, and the host’s immune system. The imbalance of these factors in genetically susceptible persons may promote a disease state. Manipulation of the intestinal microbiota with prebiotics, which can selectively stimulate the growth of beneficial bacteria, might help to maintain a healthy intestinal environment or improve a diseased one. As dietary supplements, prebiotics play an important role in maintaining environmental homeostasis in the GI tract, regulating the composition of the microbial population, and inhibiting the growth of potentially pathogenic bacteria [64,65,66].

Polysaccharide acting as prebiotics can provide a beneficial growth environment for these probiotic strains in the intestine and reduce the risk for subsequent clinical relapses of IBD [67]. Studies show that polysaccharide plays an irreplaceable role in the treatment of colitis [18,27,35,36,46,53,54,68,69]. Due to the increase in SCFA-producing bacteria, including Ruminococcus_1, and the reduction of pathogens, such as Escherichia-Shigella, in both the small intestine and cecum, Ganoderma lucidum polysaccharide was reported to be a beneficial health product for people who have intestinal dysfunction or colitis [70].

Ji et al. evaluated the modulatory effects of jujube polysaccharides (JP) on intestinal microbiota, and the influence of JP on the gut flora structure was then analyzed using an AOM/DSS-induced colitis cancer mouse model. The results indicate that the addition of JP could ward off colon cancer by ameliorating colitis cancer-induced gut dysbiosis. In addition, there was a significant decrease in *Firmicutes*/*Bacteroidetes* post-JP treatment [55].

In another study, the *Ficus carica* polysaccharide (FCP) treatment protected the goblet cells, elevated the expression of tight junction protein Claudin-1, and suppressed the formation of cytokines including TNF-α and IL-1β. The FCPS supplementation significantly reformed the gut microbiome by enhancing the abundance of S24-7, *Bacteroides*, and *Coprococus*, and suppressing the abundance of *Escherichia* and *Clostridium* at the genus level [71].

## 5. Conclusion and Future Perspectives

IBD is a chronic disease that requires constant medication. Current conventional drugs such as 5-asa, corticosteroids, immunosuppressants, and antibiotics are effective against IBD but have serious side effects and drug resistance during long-term treatment. The findings outlined in this review indicate the efficacy of polysaccharides in the treatment of IBD. In contrast to traditional drug treatments, polysaccharides come from natural products and the food we eat every day. They are mostly harmless to humans and animals. Obviously, natural polysaccharides offer great hope for the prevention and treatment of IBD. However, the mechanism of action of most polysaccharides has not been elucidated and its actual role in the treatment of IBD has not been confirmed. Further research will focus on the full explanation of the cellular and molecular mechanisms of polysaccharide action, as well as clinical trials to elucidate the efficacy and safety of different sources of polysaccharides in the treatment of IBD. More in vivo studies, especially in humans, are warranted to further elucidate and confirm the potential role of polysaccharides in IBD prevention.

## Figures and Tables

**Figure 1 molecules-27-06467-f001:**
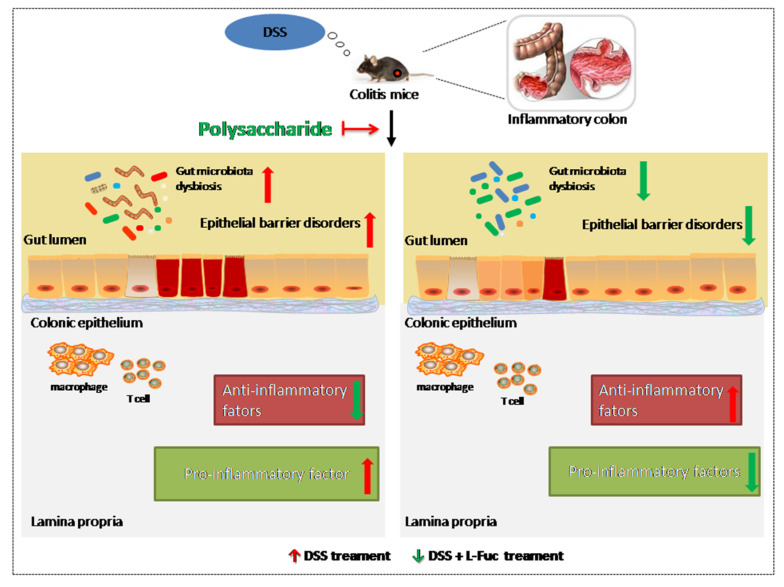
Polysaccharides can treat colitis in three aspects: regulating intestinal flora, repairing epithelial barriers, and regulating inflammatory factors.

**Table 1 molecules-27-06467-t001:** The mechanisms of different polysaccharides on experimental colitis.

Name	Resource	Models	Strain	Dose	Mechanisms	References
Polysaccharide (EP-1)	Mycelium of *Hericium erinaceus*	4% acetic acid-induced colitis	Sprague-Dawley (SD) rats	1.2 g/kg and 2.5 g/kg	the enzyme specific activity of SOD ↑, an appropriate redox balance ↑, Bcl-2 ↑, the integrity of the mitochondrial membrane ↑, the MDA content ↓, TNF-α, IL-6, IL-8 ↓, NF-κB p-65 ↓, level of ROS ↓, the caspase-3 activation ↓,	[27]
Saponins and polysaccharide	*Codonopsis pilosula* Nannf (CPN)	3% DSS-induced colitis	female C57BL/6 mice	300 mg/kg	TGF-β and IL-10 ↑, levels of acetic acid, propionic acid, butyric acid, isobutyric acid, and isovaleric acid ↑, IL-17A, IL-17F, IL-6, IL-22, and TNF-α ↓	[36]
An alkali-soluble polysaccharide (ASPP)	Purple sweet potato	DSS-induced colitis	female ICR mice	400 mg/kg	Th1, Th17 or Treg cells, acetate and propionate ↑, the SCFAs levels ↑, IL-1β, IL-6, TNF-α ↓, NF-κB ↓,	[37]
Astragalus polysaccharide (APS)	Astragalus	TNBS-induced colitis	male Sprague-Dawley rats	400 mg/kg	Treg cells ↑, STAT-5 ↑, TGF-β ↑, IL-2, IL-6, IL-17 and IL-23 ↓, ROR-gt ↓	[13]
Astragalus polysaccharide (APS)	Astragalus	3% DSS-induced colitis	male C57BL/6 mice	100 mg/kg/d, 200 mg/kg/d and 500 mg/kg/d	NLRP3, ASC, and caspase-1 ↓, IL-1β and IL-18↓, caspase-1 ↓, NLRP3 inflammasome ↓	[25]
Astragalus polysaccharide	Astragalus	TNBS-induced colitis	male Wistar rats	100 mg/kg and 200 mg/kg	NFATc4 mRNA expression ↑, TNF-a and IL-1β mRNA expressions ↓	[15]
Polysaccharide (APS)	Astragalus	TNBS-induced colitis	male SD rat	0.5g/kg/day for 14 days	IL-1β, IL-6, IL-18, TNF-α and IFN-γ, IL-10 ↑, the activation of NLRP3 inflammasome, cleavage of caspase1 ↓, β-arrestin1 expression ↓, the mRNA expressions of NLRP3, ASC, caspase1 ↓, β-arrestin1 ↓	[14]
*Rheum tanguticum* Polysaccharide	*Rheum tanguticum*	TNBS-induced colitis	Sprague-Dawley male rats	200 mg/kg/day	Prostaglandin E2 (PGE2) ↑, NF-κBp65/Lamin B1 density ratio ↓, TNF-α ↓, the COX-2/glyceraldehyde 3-phosphate dehydrogenase (GAPDH) density ratio ↓, iNOS↓	[33]
*Rheum tanguticum* polysaccharide (RTP)	*Rheum tanguticum*	TNBS -induced colitis	male Sprague-Dawley rats	200 mg/kg	CD4+T cells ↓	[20]
*Rheum tanguticum* polysaccharide (RTP)	*Rheum tanguticum*	TNBS -induced colitis	male Sprague–Dawley (SD) rats	200 mg/kg/day	IL-5 ↑, TNF-α, IFN-γ ↓, IL-4, Th1 cell cytokine ↓, Th2 cell cytokine ↑, TLR4,NF-κB/p65, p-IκBα level ↓	[19]
*Rheum tanguticum* polysaccharide (RTP)	*Rheum tanguticum*	UC induced by TNBS in BALB/c mice and CD induced by TNBS in SD rats	Adult SD rats, BALB/c mice	200 mg/kg	CD4+T cell ↓, IFN-γ ↓, Il-4 ↓,	[21]
*Rheum tanguticum* polysaccharide (RTP)	*Rheum tanguticum*	TNBS-induced colitis	male Sprague–Dawley (SD) rats	200 mg/kg/day	IFN-γ ↑	[22]
*Ganoderma lucidum* polysaccharide (GLP)	*Ganoderma lucidum*	(DSS)-induced colitis	male C57BL/6 mice	100 mg/kg	TNF-*α*, IL-1*β*, IL-6 ↓, IL-4 ↓, ROR-*γ*t ↓	[23]
Polysaccharide (MAK)	*Ganoderma lucidum* fungus mycelia	(TNBS)-induced colitis	C57BL/6(B6) mice	25 μg/mL		[24]
*Cynanchum wilfordii* Polysaccharide	*Cynanchum wilfordii*	5% DSS-induced colitis	female BALB/c mice	100 and 200 mg/kg	IL-6 ↓, TNF-*α* ↓, iNOS, COX-2 ↓, NF-*κ*B p65 ↓,	[40]
Modified apple polysaccharide (MAP)	Apple	2.5% DSS-induced colitis	male ICR mice	2.5% MAP	IL-22BP ↑, IL-22 ↓, p-STAT3, Bcl-2 and cyclin D1 ↓	[45]
β-glucan	Oat	3% DSS-induced colitis	male, ICR mice	500 mg/kg, 1000 mg/kg	TNF-α, IL-1β ↓	[18]
Oat Beta-Glucan	Oat	TNBS-induced colitis	male Sprague-Dawley rats	1% (*w*/*w*) of low molecular mass (1.7 × 106 g/mol) and high molecular mass (5.9 × 104 g/mol)	Cxcl1, Il17a, Cxcr1, Spp1 ↑, IL-6 and IL-12 ↓, TNF-α and IL-1 ↓, total cyclooxygenase (COX), prostaglandin E2 (PGE2), tromboksan A2 (TXA2), and myeloperoxidase (MPO) ↓, Ccl19, Cd40lg, Cxcr5, Il10ra, Il16, Il21, Il2rg, Il5ra, Lta, Ltb, Osm, Tnf, Tnfsf11, Tnfsf14 ↓,	[34]
*Angelica Sinensis* Polysaccharide (ASP)	*Angelica Sinensis*	2.5% DSS-induced colitis	male BABL/C mice	200 mg/kg	TJ proteins (ZO-1, occludin, and claudin-1) ↑, IL-6, IL-1b, and TNF-a ↓	[31]
*Inonotus obliquus* polysaccharide (IOP)	*Inonotus* *obliquus*	3% DSS-induced colitis	male BALB/c mice	100, 200, 300 mg/kg	Treg and Th2 ↑, Foxp3 and GATA-3 ↑, IL-4 and IL-10 ↑, p-STAT6 ↑, ROR-γt and T-bet ↓, Th17 and Th1 ↓, p-STAT1 and p-STAT3 ↓	[17]
Konjac oligosaccharide (KOS)	Konjac	TNBS-induced colitis	male SD rats,	1.0 g/kg and 4.0 g/kg	iNOS and COX-2, TNF-α and IL-1β ↓	[46]
Pulverized konjac glucomannan (PKGM)	Konjac	oxazolone-induced colitis	female C57BL/6(B6) mice	MF containing 5 % (*w*/*w*) PKGM powder, oral administration	IL-4 and IFN-γ, IL-13 ↓,	[47]
Pectic polysaccharide	*Rauwolfia verticillata* (Lour.) Baill. var. hainanensis Tsiang	DSS-induced colitis	female BALB/c mice	200 μL	Iκ Ba↑, NF- κB p65 ↓, IL-17 and TNF- a ↓	[48]
Pectic polysaccharide (PP)	*Rauvolfia verticillata* (Lour.) Baill. var. hainanensis Tsiang	(DSS)-induced colitis	female BALB/c mice	100 mg/kg	TNF-α and IL-6 ↓	[49]
Water-soluble polysaccharide (ALP-1)	*Arctium* *lappa*	DSS-induced colitis	male ICR mice	300 mg/kg	IL-10↑, IL-1β, IL-6 and TNF-α ↓	[35]
Polysaccharide	*Chrysanthemum morifolium* Ramat	DSS induced colitis	male C57BL/6 mice	75, 150, 300 mg/kg	SCFAs ↑	[50]
Polysaccharide	*Portulacae oleracea* L.	DSS-induced colitis	Kun Ming mice	0.75, 0.5, and 0.25 g/mL	PGE2 and IL-6 COX-2, STAT3 ↓	[51]
Lachnum polysaccharide (LEP)	Lachnum	2.5% DSS-induced colitis	male ICR mice	200 mg/kg	restore intestinal barrier integrity by regulating the expression of tight junction proteins and mucus layer protecting proteins,	[52]
Saponins and polysaccharides	*Codonopsis pilosula* Nannf (CPN)	3% DSS-induced colitis	female C57BL/6 mice	300 mg/kg	intestinal metabolism, recovery of the holistic gut microbiota ↑, gut microbial dysbiosis ↓	[36]
An alkali-soluble polysaccharide(ASPP)	Purple sweet potato	DSS-induced colitis	female ICR mice	400 mg/kg	regulate the composition of the gut microbiota	[37]
Yam polysaccharide and inulin	Yam	TNBS-induced colitis	SPF male Sprague-Dawley rats	300 mg/kg/day	modulate gut microbiota composition and function	[53]
Konjac oligosaccharide (KOS)	Konjac	TNBS-induced colitis	male SD rats,	1.0 g/kg and 4.0 g/kg	Bifidobacterium and Lactobacillus ↑, Escherichia coli and Enterococcus levels ↓	[46]
Water-soluble polysaccharide (ALP-1)	*Arctium* *lappa*	DSS-induced colitis	male ICR mice	300 mg/kg	Firmicutes, Ruminococcaceae, Lachnospiraceae and Lactobacillus ↑	[35]
Short-chain fructooligosaccharides(SC-FOS)	-	TNBS-induced colitis	female Wistar rats	50 g/kg	cecal lactobacilli and bifidobacteria counts ↑	[54]
*Ganoderma lucidum* polysaccharide (GLP)	*Ganoderma lucidum*	2.5% DSS-induced colitis	male Wistar rats	a basal and a GLP diet	SCFA-producing bacteria, including Ruminococcus_1, and the reduction of pathogens ↑	[55]

## Data Availability

Not applicable.

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
