# Peer review of "A Potential Role of Plant/Macrofungi/Algae-Derived Non-Starch Polysaccharide in Colitis Curing: Review of Possible Mechanisms of Action"

_molecules, 2022, doi:10.3390/molecules27196467_

Round 1
Reviewer 1 Report
I recomment to reconsider the manuscript after major revision. Below I present some comments:
L 31 Inflammatory bowel disease, also known as inflammatory bowel disease – I suppose it should be correct
L 57 “is from tens to tens of millions” of what? Da? It should be some unit here I think.
L 58 glycoside bonds – In my opinion it should be glycosidic bond
L 85 TNF-a – it should be corrected to α (alpha)
L 88 and 101 Inhibition - why with a capital letter?
L105 to prevent, L 106 maintain, regulate
L107 the sentence should not start with and
L 117 a pectic not an
L119 Erinaceus with a lowercase letter
L 127, 137 Angelica sinensis – in italics
L 150 Arctium lappa – in italics
L159 Carboxymethyl - why with a capital letter?
L165 Ulva lactuca – in italics
L156 Li W, et al. L 170Lin, H et al. – it should be unified
L 175 Chrysanthemum – in italics
Table 1 – latin names in italics
The title shuld be changed because it assumed that there are polysaccharides from plant and mushrooms investigated while there are also some examples of polysaccharides from algae.
In my opinion the manuscript need to be revised by native speaker.
The manuscript is quite chaotic so the authors should consider reorganization of the manuscript, especially in the discussion of different polysaccharides from different sources. Maybe it should be divided into plants, mushrooms, algae for example?
L262-324 This information should be provided by authors not copy from the template.
Author Response
Dear Editor and Reviewers,
Thank you for your useful comments and suggestions on our manuscript entitled "A potential role of plant/macrofungi/algae -derived non-starch poly-saccharide in colitis curing: Review of possible mechanisms of action". We have modified the manuscript according to the recommendations, and detailed corrections are listed below point by point. All revised items were highlighted in Track Changes in the new version according to reviewer’s suggestions. In addition, authors also ask for one native speaker to revise this manuscript.
Reviewers' comments and responses:
Response to comments from reviewer #1
Comment: I recomment to reconsider the manuscript after major revision. Below I present some comments:
L 31 Inflammatory bowel disease, also known as inflammatory bowel disease – I suppose it should be correct.
Response: We thank the reviewer for the comments and feedback. We revised this sentence according to the reviewer’s comments.
Comment: L 57 “is from tens to tens of millions” of what? Da? It should be some unit here I think.
Response: We thank the reviewer for the critical comments and feedback. We provided the unit of the relative molecular mass.
Comment: L 58 glycoside bonds – In my opinion it should be glycosidic bond.
Response: We thank the reviewer for the critical comments and feedback. We changed “glycoside bonds” to “glycosidic bond”.
Comment: L 85 TNF-a – it should be corrected to α (alpha)
Response: We thank the reviewer for pointing this out. We changed “TNF-a” to “TNF-α”.
Comment: L 88 and 101 Inhibition - why with a capital letter?.
Response: We appreciate for your valuable suggestion. We changed “Inhibition” to “inhibition”. We also changed “Inhibit” to “inhibit”.
Comment: L105 to prevent, L 106 maintain, regulate
Response: We thank the reviewer for this question. We revised “prevented” to “prevent”. We changed “maintained” to “maintain”. We also changed “regulated” to “regulate”.
Comment: L107 the sentence should not start with and
Response: We appreciate for your valuable and insightful suggestion. We deleted “and” in this sentence.
Comment: L 117 a pectic not an
Response: Thanks for your kind and valuable suggestion very much. We changed “an pectic” to “a pectic”.
Comment: L119 Erinaceus with a lowercase letter
Response: Thanks for your kind and valuable suggestion. We changed “Hericium Erinaceus” to “Hericium erinaceus”.
Comment: L 127, 137 Angelica sinensis – in italics.
Response: Thanks for your kind and valuable suggestion. We revised Angelica sinensis in italics.
Comment: L 150 Arctium lappa – in italics
Response: Thanks for your kind and valuable suggestion. We revised Arctium lappa in italics.
Comment: L159 Carboxymethyl - why with a capital letter?
Response: Thanks for your kind and valuable suggestion. We revised “Carboxymethyl” to “carboxymethyl”.
Comment: L165 Ulva lactuca – in italics
Response: Thanks for your kind and valuable suggestion. We revised Ulva lactuca in italics.
Comment: L156 Li W, et al. L 170Lin, H et al. – it should be unified
Response: Thanks for your kind and valuable suggestion. We unified the style in all the text.
Comment: L 175 Chrysanthemum – in italics
Response: Thanks for your kind and valuable suggestion. We revised Chrysanthemum in italics.
Comment: Table 1 – latin names in italics
Response: Thanks for your kind and valuable suggestion. We revised all the latin names in italics.
Comment: The title shuld be changed because it assumed that there are polysaccharides from plant and mushrooms investigated while there are also some examples of polysaccharides from algae.
Response: Thanks for your kind and valuable suggestion. We revised the title according to the reviewer’s comments.
Comment: In my opinion the manuscript need to be revised by native speaker.
The manuscript is quite chaotic so the authors should consider reorganization of the manuscript, especially in the discussion of different polysaccharides from different sources. Maybe it should be divided into plants, mushrooms, algae for example?
Response: Thanks for your kind and valuable suggestion. We asked for one native speaker to revise this manuscript. This review focused on the possible mechanisms of action. We divided the different function in colitis curing. We organized the discussion section as follows: Improvement of inflammatory, Recovery of epithelial barrier function and Regulation of gut microbiota.
Comment: L262-324 This information should be provided by authors not copy from the template.
Response: Thanks for your kind and valuable suggestion. We added the important information.

Reviewer 2 Report
Reviewer' comments:
This manuscript gives an overview of literature on the effects of polysaccharides on human bowel inflammatory diseases. This manuscript reviewed the potential role of polysaccharide in colitis curing. The topic was interesting. In my opinion, it can be published in this journal. Prior to be published, some minor corrections are necessary:
1. Please check the style of references.
2. Previous studies have used polysaccharide to ameliorate DSS-induced ulcerative colitis. Therefore, the section of introduction should highlight the innovation and potential application of this manuscript.
[1] Cui L, Guan X, Ding W, Luo Y, Wang W, Bu W, Song J, Tan X, Sun E, Ning Q, Liu G, Jia X, Feng L. Scutellaria baicalensis Georgi polysaccharide ameliorates DSS-induced ulcerative colitis by improving intestinal barrier function and modulating gut microbiota. Int J Biol Macromol. 2021 Jan 1;166:1035-1045. doi: 10.1016/j.ijbiomac.2020.10.259.
[2] Zou Q , Zhang X , Liu X , Li Y , Tan Q , Dan Q , Yuan T , Liu X , Liu RH , Liu Z . Ficus carica polysaccharide attenuates DSS-induced ulcerative colitis in C57BL/6 mice. Food Funct. 2020 Jul 1;11(7):6666-6679. doi: 10.1039/d0fo01162b.
3. In second paragraph of 2. Improvement of inflammatory, “TNF-a and IL-1b” should be “TNF-α and IL-1β”.
4. In third paragraph of 2. Improvement of inflammatory, “Rheum tanguticum polysaccharide (RTP) have….” should be “Rheum tanguticum polysaccharide (RTP) has”.
5. In fourth paragraph of 2. Improvement of inflammatory, “Ganoderma lucidum polysaccharides (GLP) was…” should be “Ganoderma lucidum polysaccharides (GLP) were…”.
6. In fifth paragraph of 2. Improvement of inflammatory, “Oxidative stress signaling is involved in and promote” should be “Oxidative stress signaling is involved in and promoted”.
Author Response
Dear Editor and Reviewers,
Thank you for your useful comments and suggestions on our manuscript entitled "A potential role of plant/macrofungi/algae -derived non-starch poly-saccharide in colitis curing: Review of possible mechanisms of action". We have modified the manuscript according to the recommendations, and detailed corrections are listed below point by point. All revised items were highlighted in Track Changes in the new version according to reviewer’s suggestions. In addition, authors also ask for one native speaker to revise this manuscript.
Reviewers' comments and responses:
Response to comments from reviewer #2
Comment: This manuscript gives an overview of literature on the effects of polysaccharides on human bowel inflammatory diseases. This manuscript reviewed the potential role of polysaccharide in colitis curing. The topic was interesting. In my opinion, it can be published in this journal. Prior to be published, some minor corrections are necessary:
- Please check the style of references.
Response: We thank the reviewer for pointing this out. We double-checked all the reference style.
Comment: 2. Previous studies have used polysaccharide to ameliorate DSS-induced ulcerative colitis. Therefore, the section of introduction should highlight the innovation and potential application of this manuscript.
[1] Cui L, Guan X, Ding W, Luo Y, Wang W, Bu W, Song J, Tan X, Sun E, Ning Q, Liu G, Jia X, Feng L. Scutellaria baicalensis Georgi polysaccharide ameliorates DSS-induced ulcerative colitis by improving intestinal barrier function and modulating gut microbiota. Int J Biol Macromol. 2021 Jan 1;166:1035-1045. doi: 10.1016/j.ijbiomac.2020.10.259.
[2] Zou Q , Zhang X , Liu X , Li Y , Tan Q , Dan Q , Yuan T , Liu X , Liu RH , Liu Z . Ficus carica polysaccharide attenuates DSS-induced ulcerative colitis in C57BL/6 mice. Food Funct. 2020 Jul 1;11(7):6666-6679. doi: 10.1039/d0fo01162b..
Response: We thank the reviewer for this suggestion. We added some important and useful references to improve the introduction section.
Comment: 3. In second paragraph of 2. Improvement of inflammatory, “TNF-a and IL-1b” should be “TNF-α and IL-1β”.
Response: We would like to thank reviewer for their insightful comments to improve our manuscript. We changed “TNF-a and IL-1b” to “TNF-α and IL-1β”.
Comment: 4. In third paragraph of 2. Improvement of inflammatory, “Rheum tanguticum polysaccharide (RTP) have….” should be “Rheum tanguticum polysaccharide (RTP) has”.
Response: Thank you very much for your comments. We revised “have” to “has”.
Comment: 5. In fourth paragraph of 2. Improvement of inflammatory, “Ganoderma lucidum polysaccharides (GLP) was…” should be “Ganoderma lucidum polysaccharides (GLP) were…”.
Response: We thank the reviewer for the critical comments and feedback. We revised “was” to “were”.
Comment: 6. In fifth paragraph of 2. Improvement of inflammatory, “Oxidative stress signaling is involved in and promote” should be “Oxidative stress signaling is involved in and promoted”.
Response: We thank the reviewer for raising the important point. We changed “promote” to “promoted”.
Bin Du
Hebei Key Laboratory of Natural Products Activity Components and Function, Hebei Normal University of Science and Technology, Qinhuangdao, Hebei 066004, China; bindufood@aliyun.com

Round 2
Reviewer 1 Report
I accept inroduced corrections.
Malva sylvestris in table should be in italics